# Thermography as a Method for Bedside Monitoring of Infantile Hemangiomas

**DOI:** 10.3390/cancers14215392

**Published:** 2022-11-01

**Authors:** Juan Antonio Leñero-Bardallo, Begoña Acha, Carmen Serrano, José Antonio Pérez-Carrasco, Juan Ortiz-Álvarez, José Bernabéu-Wittel

**Affiliations:** 1Department of Electronics and Electromagnetism, Universidad de Sevilla, Av. Reina Mercedes s/n, 41012 Seville, Spain; 2Department of Signal Theory and Communications, Universidad de Sevilla, Escuela Técnica Superior de Ingeniería, Camino de los Descubrimientos s/n, 41092 Seville, Spain; 3Dermatology Unit, Virgen de Rocío University Hospital, Av. Manuel Siurot, s/n, 41013 Seville, Spain

**Keywords:** imaging, infantile hemangioma, thermography

## Abstract

**Simple Summary:**

Infantile hemangiomas are the most frequent vascular tumors in infants. Conventional diagnosis relies on visual observation of vascular anomalies and ancillary techniques such as Doppler ultrasonography or magnetic resonance. These diagnostic methods are not straightforward and may require trained operators. Children are usually apprehensive of them because they may require the use of gels, general anesthesia, and contact with their skin. As temperature variations induced by a hemangioma can be measured very accurately, we explored thermographic imaging as an alternative tool for clinically monitoring hemangioma extensions before and during treatment. We found a correspondence between the satisfactory evolution of hemangiomas during treatment, their extension, and their temperature.

**Abstract:**

Infantile hemangiomas occur in 3 to 10% of infants. To predict the clinical course and counsel on treatment, it is crucial to accurately determine the hemangiomas’ extension, volume, and location. However, this can represent a challenge because hemangiomas may present irregular patterns or be covered by hair, or their depth may be difficult to estimate. Diagnosis is commonly made by clinical inspection and palpation, with physicians basing their diagnoses on visual characteristics such as area, texture, and color. Doppler ultrasonography or magnetic resonance imaging are normally used to estimate depth or to confirm difficult assessments. This paper presents an alternative diagnosis tool—thermography—as a useful, immediate means of carrying out accurate hemangioma examinations. We conducted a study analyzing infantile hemangiomas with a custom thermographic system. In the first phase of the study, 55 hemangiomas of previously diagnosed patients were analyzed with a thermal camera over several sessions. An average temperature variation before and after treatment of −0.19 °C was measured. In the second phase, we selected nine patients and assessed their evolution over nine months by analyzing their thermographic images and implementing dedicated image processing algorithms. In all cases, we found that the thermal image analysis concurred with the independent diagnoses of two dermatologists. We concluded that a higher temperature inside the tumor in the follow-up was indicative of an undesirable evolution.

## 1. Introduction

Infantile hemangiomas (IHs) are benign tumors produced by clonal expansions of endothelial cells. They occur in 3% to 10% of infants [1], appearing at birth or within the first 4 to 6 weeks after birth [2,3,4]. If they require treatment, oral propranolol is currently the most common option [5]. The evaluation of IH extension and volume is not straightforward. Hemangiomas may present irregular patterns or be covered by hair, or they may not be directly accessible for visual inspection. Léauté-Labrèze et al. monitored IH evolution using standardized photographs evaluated by different experts [1]. In practical clinical situations, physicians monitor IH evolution by visually inspecting earlier photographs that may not show the real boundaries of the tumor.

The conventional diagnostic techniques for assisting practitioners in determining IH extension and localization are Doppler ultrasonography and magnetic resonance imaging (MRI). These diagnostic methods are not straightforward, may not always be available in the first consultation, and may require trained operators. In the case of MRI for infants, anesthesia is required.

During the last decade, infrared cameras have fallen in price and have become affordable, reliable clinical diagnosis tools [6]. They are currently in very high demand because they are being massively employed in public environments to identify people who may be affected by COVID-19 [7]. Thermal cameras can accurately measure spatial body temperature variations [8,9,10,11,12,13] inside hemangiomas [11,12], vascular malformations [14,15], and other clinical entities [8,10].

As an alternative to ancillary diagnostic techniques, many authors have already described and explored the great potential of thermographic images for studying and classifying certain types of vascular anomalies [16,17] that produce local alterations in body temperatures [8,9,10,11,12,13,14,15,16,17,18]. IHs are just one of a vast spectrum of vascular anomalies that can be easily detected using thermography precisely, because they induce body temperature variations detectable by IR cameras [11,12,15]. According to Zeroth and the First Laws of Thermodynamics, there is a heat transfer to the environment due to the movement of a fluid (the blood) by the convection mechanism. Therefore, if the hemangioma increases the local blood vessels, the local temperature is consequently increased. In 2016, Burkes et al. [11] studied the dynamic thermal response of IHs to temperature variations and concluded that dynamic temperature behavior in IHs is different from that in bare skin. In 2017, Strumila et al. published a study presenting thermography as a tool for accurately predicting the evolution of an untreated hemangioma [18]. All of these earlier works relied on commercial systems that simply executed standard image processing routines, incorporating peripherals to expand their capabilities. Such systems were not, therefore, customized for the clinical analysis of IHs.

To the best of our knowledge, no standardized, automated procedures exist for assessing the evolution and extension of IHs by analyzing thermographic images. To extend the clinical use of IR cameras, it is necessary to define which image features are relevant for IH diagnosis, establish reference values for those parameters to assess the IH’s evolution in the follow-up, and present algorithms for processing IR images to extract the relevant data.

## 2. Materials and Methods

### 2.1. Materials

A custom system [15] was built to acquire and process medical images in both the visible and infrared (IR) bands. For a detailed description of the system’s implementation and capabilities, we refer the reader to an earlier study [15]. Unlike some non-customizable commercial devices, this system can easily be adapted to different application scenarios by adding new peripherals and/or custom image processing algorithms. Portable, autonomous, and easy to use, it offers immediate results at the moment of examination and has a competitive cost. The system comprises the following elements (see Figure 1):A single board computer (SBC) is the core of the system. We chose a Raspberry Pi IV board to build the prototype. It controls all of the peripherals and sensors that make up the system. It has a Linux-based operating system and can run real-time image processing algorithms. A custom user interface was programed to acquire, display, and process the images. SBC also has Wi-Fi and Ethernet connectivity, and an expansion bus is available to add extra peripherals or sensors if necessary.A visible spectrum camera based on a Sony IMX219 8-megapixel sensor.A LEPTON thermal camera operating the long wave infrared (LWIR) band. After calibration, it can take thermal images and measure absolute temperature values within the visual scene.A Melexis MLX90614D IR thermometer to calibrate the thermal camera.A 5-INCH HDMI LCD Waveshare touch screen to display the images captured by the two image sensors and to control the user interface. The interface allows for the simultaneous acquisition of visible and IR images.A standard Poweradd 5 V battery to make the system autonomous and portable.

### 2.2. Methods

A pilot study was conducted in two phases. Figure 2 shows a flowchart summarizing the steps and phases in this study. In a first phase, 55 patients with at least one IH were examined using thermography in their dermatology consultation. The patients were diagnosed beforehand by two independent dermatologists using conventional diagnostic methods. The inclusion criteria were as follows:Participants had to have at least one IH diagnosed simultaneously and independently by the two specialists. Patients with vascular anomalies that were not identified as hemangiomas were excluded from the study.The patients’ IHs must not have been treated previously.All variants of IHs were accepted for the study, independently of their proliferation phase or the patient’s medical history, gender, or ethnicity.There was no age limit for patients eligible to participate in the study.Patients’ guardians had to agree for their charges to be subjected to thermographic analysis in all upcoming clinical sessions. The scheduled periodicity between control sessions was three months.

Patients were continuously recruited for two years, from May 2019 to May 2021. After recruitment, each patient received personalized treatment appropriate to their type of IH, its proliferation phase and localization, their age, weight, etc. Sessions were spaced three months apart in time.

For each patient, two different images (clinical and infrared) were acquired simultaneously (following a specific protocol: At a fixed distance of 30 cm, and under the same light conditions and ambient temperature). In each session, the infrared camera was calibrated with a thermometer to translate radiation levels in the LWIR band into absolute temperatures. The temperature variation between the center of the IH and the surrounding healthy skin was calculated in each session. Leñero-Bardallo et al. reported a correlation between temperature variations and high or low blood flow in vascular malformations [19]. The evolution of this temperature variation was analyzed for each patient during the course of the three sessions. Figure 3 graphically illustrates how the system was used to study one IH.

In the second phase of the study, the evolution of IHs was analyzed in 9 of the 55 patients. The inclusion criterion for this second phase was to select patients who had been examined with the IR camera at least three times during their nine months of dermatology consultations. The first phase had a time span of two years. Patients whose first visit was close to the end of the study were thus not eligible for the second phase. Unfortunately, for several unexpected reasons, some patients were discarded in the second phase of the pilot study because they could not attend three sessions in a row as scheduled. This second phase aimed to determine whether the hemangioma’s evolution using different medications correlated with the thermal analysis provided by the camera. The eligibility criterion for the second phase was strict because it was important to acquire enough clinical records and experimental data to determine whether the thermographic analysis and the independent clinical evaluation for each IH concurred.

#### 2.2.1. Image Processing User Interface

A custom interface (shown in Figure 1b) was implemented to display and process the medical images acquired during the clinical examination. The user can define a custom image processing algorithm that can be executed in real-time. These algorithms facilitate image interpretation and, consequently, diagnosis. We chose the C++ programming language to implement the user interface and the image processing algorithms described in the article. The C++ language is compatible with the implementation of real-time execution algorithms. Another reason for choosing C++ was that there is free access to the public OpenCV repository [20]. The following sections describe the image processing functions that were implemented.

#### 2.2.2. Image Segmentation

The region of interest for the thermographic analysis is the skin. It is sometimes convenient to segment the skin from the patient’s clothes, ornaments, or body regions totally covered by hair because these artifacts may report erroneous thermal information. To do this, a thresholding algorithm removes all the regions in the thermal image with a temperature below a pre-established threshold. The thresholding algorithm used was Otsu’s method [21,22,23], which automatically determines the optimum threshold to be applied to an image with an approximate bimodal distribution. The hemangioma produces an increase in the local skin temperature that corresponds to the histogram mode with the highest temperature. To use Otsu’s method, the algorithm should ideally be applied to a histogram with a bimodal distribution. The scene’s background creates a low temperature histogram mode that can be easily removed by removing all of the inert objects from the image that are not compatible with body temperature. All of the temperature values below 34 °C, for instance, can be discarded. After performing this operation, Otsu’s method can be applied to segment the skin. The processing steps involved in skin segmentation in the thermal image are detailed below:Histogram counts are computed for the thermal image.Otsu’s method is applied using the histogram counts to calculate the optimum threshold temperature value, *TH*, to segment the image.The image is thresholded: pixel values lower than *TH* are set to zero.The resulting pixel matrix is stored in the memory and displayed on the system screen.

#### 2.2.3. Temperature Contour Lines and Area Calculations

The calculation of an IH area can be automated to provide an objective reference value that indicates whether the administrated dose should be maintained, increased, or decreased. The contour line area estimation procedure is as follows:Compute and plot *N* temperature contour lines within the [35.5,38]*°C* range, that is to say, the range in which the temperature usually varies with the presence of vascular anomalies. This was done by selecting the *findContours()* predefined contour extraction routine from the OpenCV library [20].Select a temperature value to delimit the malformation boundary. The same value will be used to compare the area of the affected region in the different clinical sessions. The *findContours()* function provides coordinate pairs $P_{i}\left(x_{i},y_{i}\right)\mathrm{with}i=1,\;\ldots \;,k$
of the *k* vertices that delimited the contour line. Note that the vertices have to create a closed contour. The start and end point vertex coordinates are the same and must therefore be considered twice in the computation, i.e., $P_{1}\left(x_{1},y_{1}\right)=P_{k}\left(x_{k},y_{k}\right)$.
The corresponding contour line is plotted separately for a detailed analysis.The extension (the number of pixels corresponding to the location of the vascular malformation) is computed using the coordinate method [21,22], in two steps:


(1)
$$S_{1}=x_{1}y_{2}+\sum_{i=1}^{N-1}x_{i}y_{i+1},\;S_{2}=x_{2}y_{1}+\sum_{i=1}^{N-1}x_{i+1}y_{i}\;$$



(2)
$$A=\frac{\left|S_{1}-S_{2}\right|}{2}$$


This area value, *A*, is expressed in pixel units. It represents the region of interest inside the pixel array that corresponds to the IH location, and it can be taken as a reference to assess the patient’s prognosis in the follow-up. An increment in the parameter is equivalent to a regrowth of the IH. The system can give the absolute IH area, *A_abs_*[cm^2^], by considering the distance from the patient’s malformation to the sensor, *d*, the pixels’ physical dimensions (*W*[cm^2^] and *L*[cm^2^] width and length, respectively), and the optical focal distance, *f*, i.e.,


(3)
$$A_{abs}\left[{\mathrm{cm}}^{2}\right]=A^{\prime }\cdot {\left(\frac{f}{d}\right)}^{2}\cdot W\left[{\mathrm{cm}}^{2}\right]\cdot L\left[{\mathrm{cm}}^{2}\right]$$


## 3. Results

### 3.1. Medical Image Analysis and Processing

Figure 4a–f shows a comparative analysis of a hemangioma that evolved over five months, before and after treatment. The images on the top row were taken before starting the treatment. The patient was 43 days old at the first consultation. The images on the bottom row were obtained after initiating the treatment. Figure 4a,d shows the tumor in the visible spectrum. Figure 4b,e shows the same image in the LWIR band. Figure 4c,f displays the histograms of the infrared images. Each histogram bin represents the number of pixels with a certain temperature value. The temperature contrast between the region affected by the IH and the surrounding healthy skin was higher before starting the treatment. According to the image analysis, the extension of the IH decreased after administrating oral propranolol. The area is thus indicative of the IH’s involution [4,24,25]. The procedure can also be used to calculate an IH’s extension when slightly covered by hair, as shown in Figure 4d. Histograms are useful for identifying the number of pixels affected by the tumor (extension) from different image regions. Body regions not affected by the IH and the image background have lower temperatures. The histogram modes show the average temperatures of each region. The one associated with the IH region is less prominent after treatment, with the body temperature distribution becoming more uniform.

Similarly, Figure 5a–f illustrates the evolution over two months of an IH located in the patient’s back, before treatment (top row) and after treatment (bottom row). The patient was 3 days old at the first consultation. In this case, the area of the affected region can also be seen to have gotten smaller. There is also a temperature reduction between the affected and the non-affected region. IH involution is thus associated with decreases in area and temperature.

### 3.2. Image Segmentation, Area Calculation, and Contour Extraction Results

Figure 6 illustrates the thresholding and contour line extraction operations. Figure 6b,c shows the IR image in Figure 6a after thresholding. Figure 6d,e shows the temperature contour lines corresponding to Figure 6a. The user can zoom in on a particular temperature contour line to accurately locate the hemangioma and determine its extension. The hemangioma extension is usually larger than that perceived by the practitioner in a visual inspection based on texture or color variations inside the region affected by the IH.

Figure 7 illustrates one particular example of the usefulness of the area estimation method. In this case, the patient has a deep focal IH. Area estimation by visual inspection is challenging because some inner parts of the IH are not directly visible. However, the thermal image provides more accurate information about the IH’s extension.

### 3.3. Pilot Study

The goal of the two-year pilot study was to validate the usefulness of thermographic images in determining the extension of IHs and assessing their satisfactory evolution during treatment. The research study phases and steps are shown in the flow chart in Figure 2. During the consultations, thermographic analyses were compared with the previous diagnoses of two dermatologists. In the first stage, the IHs of 55 patients were studied. The patients had been diagnosed with different kinds of hemangiomas [26,27,28], including congenital and infantile hemangiomas. In terms of spatial distribution, the IHs were focal or segmental and their depths were superficial, deep, or mixed. Abortive hemangiomas were also considered in the study. In all cases, it was the patient’s first visit to the dermatologist. The degree of proliferation of the IHs varied. Table 1 summarizes the experimental data obtained after the first visit. In patients with multiple IHs, only the largest one was considered; 87.3% of the IHs induced temperature variations affecting the surrounding skin. Temperatures in this initial analysis varied greatly because they were strongly dependent on the hemangioma subtype [26,27,28], its proliferation phase, and the time that had elapsed between its onset and the first clinical inspection. The average temperature variation was 0.2 °C. The 90% confidence interval for the temperature variation was [0.16, 0.25] °C.

In the second phase of the study, nine patients with high or moderate risk IHs were selected from the 55 patients under study. These patients underwent different treatments (oral propranolol, oral nadolol, topical timolol, or just watch-and-wait to see how the IH evolved) and were monitored with our thermal camera for nine months. Some of them had more than one IH and one patient presented multifocal cutaneous hemangioma. In total, there were 17 IHs under study in this second round. IH temperatures were monitored during three sessions spaced three months apart in time. The results obtained in the second phase are summarized in Table 2. In the first session, the average IH temperature variation between the IH and its surrounding healthy skin was 0.38 °C. In the last session, the average variation was 0.27 °C. The sixth column shows the degree of agreement between temperature variation and the dermatologists’ treatment decision made using ancillary techniques (Doppler ultrasonography, visual inspection, and magnetic resonance) [4,24]. We considered that there was agreement between the two if:(i)There was a relative temperature increment in the hemangioma over time and the dermatologists diagnosed a negative response to treatment.(ii)There was a relative temperature decrement in the hemangioma, and the dermatologists diagnosed a positive response to treatment.(iii)There was a null temperature increment in the hemangioma, but the IH area estimation provided by the system was substantially lower than in the previous session.

Analyzing the results, we can conclude that in all cases, there was agreement between the results provided by the thermographic analysis and the dermatologists’ diagnoses. The last column in Table 2 shows the average value for the IH temperature variations during treatment. Overall, we measured an average IH temperature variation in the surrounding skin of −0.19 °C before and after treatment, indicating that temperature decrements are associated with a correct evolution.

For a deeper analysis and understanding of the results and the IH evolution in the second phase, Table 3 and Table 4 provide a detailed summary of how the 17 IHs of the 9 patients recruited after treatment evolved. Table 3 shows the clinical and demographic characteristics of the nine patients, while Table 4 summarizes temperature variations, response to treatment, incidences, and side effects. The IH dimensions shown in the fourth column correspond to the first session and were measured with a meter tape. Relative area increments/decrements between sessions were monitored with the IR system. The IH temperatures showed a negative variation during the course of the three sessions, accompanied by an IH area reduction during treatment in seven of the nine patients. Three patients showed a transitory positive temperature variation after discontinuation of the treatment. This is concordant with IH regrowth after stopping treatment. In one of these three cases, the IH regrowth was accompanied by ulceration. Local ulceration contributes to a rise in local body temperatures, because blood or blood vessels are directly accessible in such regions. For all 17 IHs under study, there was thus agreement between the thermographic analysis and the response to treatment.

## 4. Discussion

Infrared cameras are helpful tools for monitoring IH temperature activity and extension during treatment. The experimental results obtained from 55 patients in their first dermatology consultation show that there is a temperature increase in IHs compared with healthy skin. This was the case in 87.3% of the 55 cases. The IR sensor selected for this study was only able to detect temperature variations larger than 0.1 °C, so in many of the cases where temperature variations were not detected, this was due to the sensor’s limited temperature resolution. It should be noted that thermography always detected IHs in the proliferative phase. The sensor’s temperature resolution limited the analysis of IHs in an involutive stage. The second phase of the study, conducted with nine patients over nine months, showed agreement between the evolution of clinically treated IHs and their variations in temperature and extension. The treatment response assessment of two independent dermatologists and the thermographic data analysis concurred in all nine patients who participated in the second phase.

In contrast with previous studies of IH thermal activity [14,18], this is the first research work to focus on the analysis and processing of thermographic images as a means of assessing a hemangioma’s evolution. Early works on thermography focused on demonstrating that IHs alter temperatures locally, but did not provide accurate temperature information. Such temperature variations were, however, detected with IR thermometers or infrared cameras [8,13,29]. Some subsequent works analyzed the average temperature of a thermographic image as a whole [11,12,30], but did not describe how the IH could be segmented from the healthy skin or other scene elements (hair, clothes, objects, etc.) that are inevitably present in the thermal image and which alter temperature measurements. The second phase of this study, the assessment of hemangioma evolution over time comparing conventional methods with thermographic images, represented a novelty over prior research works. We identified the image features that are relevant for assessing treatment objectively. We also obtained insights about the system’s implementation and how to customize it and develop image processing algorithms to extract relevant information about IH evolution. Earlier research into thermography used non-customizable commercial infrared cameras, with no possibility of incorporating image processing algorithms adapted to specific application scenarios. Moreover, the medical applicability of standard infrared cameras is limited due to their reduced capability to coexist and exchange data with other peripherals.

Thermography can coexist with ancillary diagnostic techniques based on Doppler ultrasound analysis or magnetic resonance. Thermographic analysis is immediate, non-invasive, and capable of estimating hemangioma extension when tumors are partially covered by hair and do not have a uniform color or texture. The rapid spread and development of infrared cameras have made this technology easily accessible for clinicians. During the last ten years, the price of IR sensors has dropped significantly, making them affordable and accessible for a broader range of applications [8,15]. IR sensors are now massively deployed in public environments to detect people affected by COVID-19. The tendency is to reduce their cost even more, increase their pixel numbers and improve their specifications. There is, thus, a need for research to assess the utility of these emerging devices in unconventional medical application scenarios. Infrared cameras have the potential to become an excellent complementary diagnostic tool for dermatologists. In previous work, we demonstrated their ability to differentiate between high- and low-flow vascular anomalies [19]. In this contribution, we have shed light on how they can assist in the evaluation of IH treatment. The main limitation in our pilot study was the low number of patients analyzed in the second phase. In future research, we plan to create a public database with a larger number of thermal images showing different IH types in their respective evolutionary phases during treatment. The database will help us find out more about how to correlate thermal image analysis with the evolutionary assessment of vascular anomalies. We also plan to explore how the proposed system, image processing algorithms, and medical image acquisition protocols can be used to study other vascular anomalies that alter local body temperatures. One area of interest is the use of temperature analysis to differentiate lymphatic and vascular malformations. As lymphatic malformations induce temperature decrements in the regions affected, thermography can be employed to determine whether there is a lymphatic component in a given vascular anomaly.

### Method Limitations and Considerations

To take accurate absolute temperature measurements between the different sessions, it was necessary to acquire the thermographic images with the IH at a fixed distance from the infrared sensor. The thermographic device must be calibrated between sessions if the room temperature is not regulated. The calibration procedure is automated and can be performed at the beginning of the session if necessary. Absolute temperature values are not required to determine whether an IH’s extension is changing during treatment. Growth can be monitored between sessions by comparing the relative temperature difference between the IH region and the surrounding skin. For precise temperature measurement using a thermal camera, the hemangioma must be directly exposed to the camera’s field of view. Hair or other elements in between the camera and the IH alter the temperature measurement.

One limitation of the study is that, although the correlation between temperature variation and response to treatment has been shown, the correlation between temperature variation and blood flow alteration has not been evaluated. This correlation between blood flow and response to treatment is still in debate [30].

## 5. Conclusions

This study offers insights into the use of thermographic cameras to diagnose and monitor the evolution of infantile hemangiomas. Using a custom thermographic system, we shed light on two important aspects of the use of thermal imaging to study IHs. On the one hand, we explain in detail how IR images can be processed to obtain relevant information in this application scenario (i.e., to compute hemangiomas’ extensions and the temperature variations they induce). We also provide experimental data, after having analyzed 55 patients before and after treatment. On the other hand, the results of a pilot study correlating thermographic image analysis with independent medical assessments show agreement between the satisfactory evolution of infantile hemangiomas and their extension and temperature. According to the analysis of 19 infantile hemangiomas, higher temperatures and area regrowth are indicative of hemangioma involution. An IH’s area and temperature can be accurately measured and clinically monitored using infrared cameras. IR cameras are powerful tools for dermatologists. They can potentially detect body temperature variations induced by any medical entity. At the same time, they are now being massively deployed in public environments, leading to lower prices and a better performance at a system level. There is much scope for research and for the ongoing customization of systems through the incorporation of dedicated, embedded image processing algorithms that can be adapted to extract the most relevant image features in each application scenario.

## Figures and Tables

**Figure 1 cancers-14-05392-f001:**
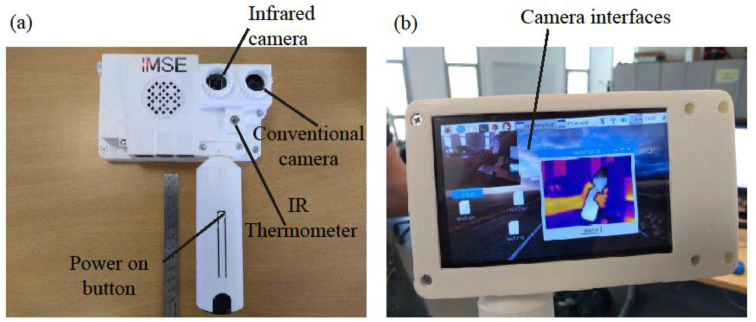
Details of system implementation: (**a**) front view; (**b**) back view. The system has two cameras running simultaneously. One operates in the visible spectrum and the other in the infrared band. The second camera can measure absolute temperature values after calibration. An LCD display is available to visualize data from the sensors.

**Figure 2 cancers-14-05392-f002:**
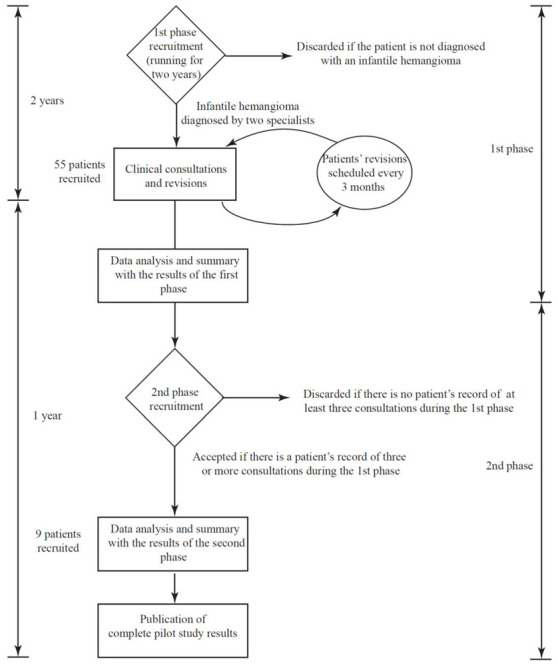
Flow chart summarizing the different steps and phases involved in the pilot study.

**Figure 3 cancers-14-05392-f003:**
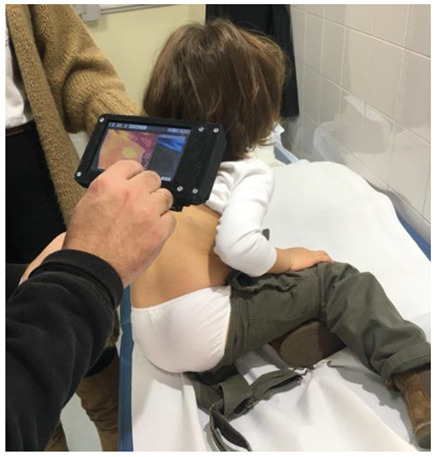
Illustration of how the system is used to study an IH. The IH is analyzed without touching the patient. The distance between the IH and the system is constant. On the screen, a real-time thermographic image is displayed. This photograph corresponds to a yearly control session for a previously treated IH patient who did not participate in the trial study.

**Figure 4 cancers-14-05392-f004:**
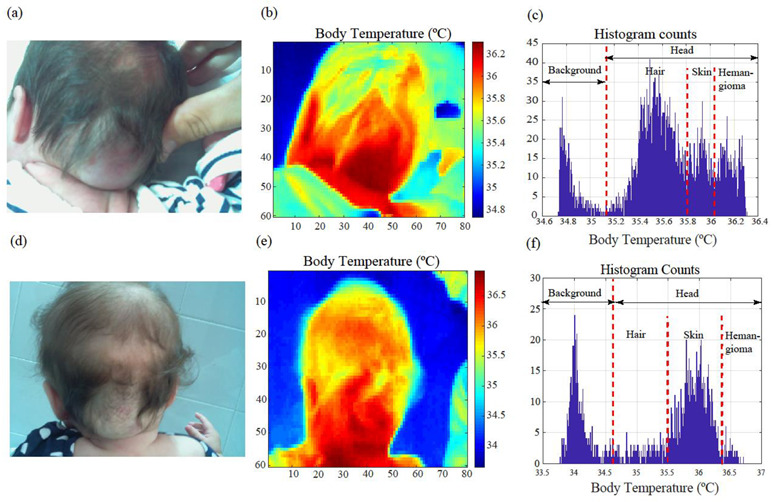
(**a**–**f**) Study of the temperature of an infantile hemangioma in the occipital region before and after five months of treatment with oral propranolol. The patient was 43 days old at the first consultation. (**a**,**d**) Images of the hemangioma obtained with a standard digital camera. (**b**,**e**) Thermographic images of the same hemangioma displaying absolute temperature values. A temperature increment in the hemangioma compared with its surroundings is observed. (**c**,**f**) Histograms of the thermal images. The histogram modes correspond to the average temperature values in the different scene regions, i.e., image background, hair, skin, and hemangioma.

**Figure 5 cancers-14-05392-f005:**
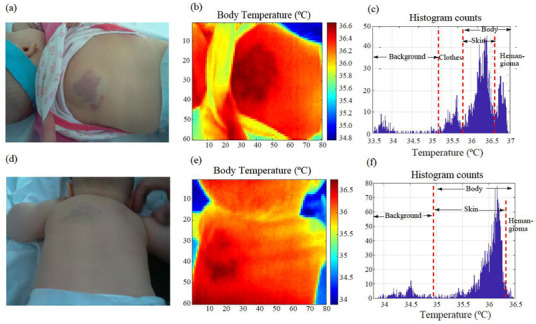
(**a**–**f**) Study of the temperature of an infantile hemangioma on the patient’s back, before and after a two-month course of treatment with oral propranolol. The patient was 3 days old at the first consultation. (**a**,**d**) Images of the hemangioma obtained with a standard digital camera. (**b**,**e**) Thermographic images of the same hemangioma, displaying the absolute temperature values. A temperature increment in the hemangioma compared with its surroundings is observed, (**c**,**f**) Histograms of the thermal images. The histogram modes correspond to the average temperature values in the different scene regions, i.e., image background, hair, skin, and hemangioma.

**Figure 6 cancers-14-05392-f006:**
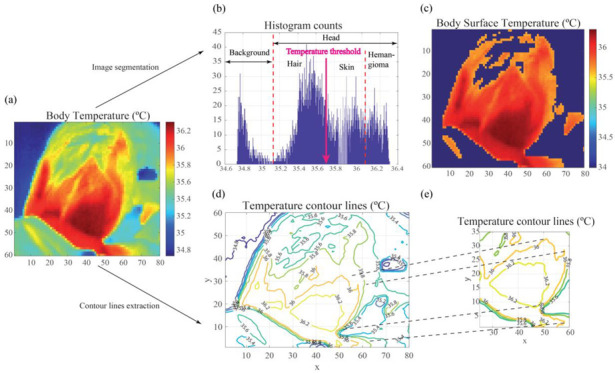
Illustration of the processing steps involved in image segmentation. (**a**) Original thermographic image with absolute temperature values. (**b**) Histogram of the original thermographic image. The temperature threshold, *TH*, determined by Otsu’s algorithm for segmenting skin, is highlighted. (**c**) Segmented image displaying only the temperature values of the patient’s skin. (**d**) Temperature contour lines extracted from the original image. The contour lines with the highest temperature correspond to the IH’s location. (**e**) Detail of the temperature contour lines in the region affected by the malformation. The area inside a temperature contour line can be computed using the coordinate method.

**Figure 7 cancers-14-05392-f007:**
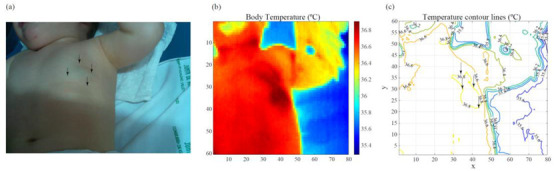
Illustration of the applicability of the area estimation method for a deep focal IH. Area calculation is challenging by visual inspection. Thermography facilitates the process. (**a**) Visible image. (**b**) Thermographic image. (**c**) Detail of the temperature contour lines. The arrows indicate the region affected by the IH.

**Table 1 cancers-14-05392-t001:** Summary of the clinical experience analyzing infantile hemangiomas with a thermal camera (first phase of the study). Preliminary results before initiating treatment.

No. of Patients	No. of Hemangiomas Studied (First Visit Only)	No. of Images withTemperature Variation	Average Temperature Increment (∆°C)	90% Confidence Interval for Temperature Variation (∆°C) [Min, Max]	Percentage of Hemangiomas with Temperature Variations
55	55	48	0.20	[0.16, 0.25]	87.3%

**Table 2 cancers-14-05392-t002:** Summary of the clinical experience analyzing infantile hemangiomas with a thermal camera (second phase of the study). Results obtained after examining nine patients in three sessions during their nine months of treatment.

No. of Patients	No. of Hemangiomas Studied (Several Visits)	Positive ∆°C	Negative ∆°C	Null ∆°C	Agreement with Dermatologists’ Decision	Average Initial/Final Temperature Variation Induced by Hemangioma in the Surrounding Skin ${\bar{\Delta T}}_{initial},\;{\bar{\Delta T}}_{final}$	Average Hemangioma Temperature Variation during Treatment (∆°C)
9	17	2	14	1	17/17	0.38/0.27	−0.19

**Table 3 cancers-14-05392-t003:** Clinical and demographic characteristics of the nine patients with IHs monitored using thermography.

Case	Sex	Age (Years)	IH Type	Localization and Dimensions (mm)	Age at First Consultation (Months)	Treatment Duration (Months)	Administered Treatments
1	F	2	Mixed focal	Dorsal 40 × 40	1.5	Ongoing	Oral propranolol, oral nadolol
2	F	2	Superficial focal	Scalp 30 × 30	2	10	Topical timolol
3	F	5	Mixed focal	Right eye inner canthus-	1.5	24	Oral propranolol, oral nadolol
4	M	1	Segmental	Perianal Ø150	1	11	Watch and wait, oral propranolol
5	M	2	Deep focal	Pectoral 30 × 30	6	Ongoing	Topical timolol
			Mixed focal	Lumbar 60 × 40			
6	F	2	Multifocal cutaneous hemangiomas	Scalp (2), left mandibular ramus (1), dorsal (2), right elbow (1), left temple (1), 12 × 2 (max. size)	3	Ongoing	Topical timolol, oral propranolol
7	F	2	Superficial focal	Occipital 55 × 24	0.5	6	Watch-and-wait
8	F	1	Superficial focal	Left forearm 10 × 7	0.5	12	Topical timolol
9	F	1	Superficial focal	Vertex 8 × 10	1	Ongoing	Topical timolol, oral propranolol
			Superficial focal	Left scapula 11 × 9			

**Table 4 cancers-14-05392-t004:** Comparative analysis between the thermographic analysis and the response to treatment in the nine patients with IHs monitored using thermography. Temperature analysis and response to treatment concurred in all cases.

Case	Hemangioma Temperature Variation	Clinical Response	Incidences and Side Effects
1	Overall negative increment. Positive increment after treatment discontinuation (2nd month)	Flattening, disappearance of erythema	Growth (3rd month) upon early treatment discontinuation (2nd month)
2	Negative increment	Total disappearance	No
3	Overall negative increment. Positive increment detected after treatment discontinuation (1st month)	Total disappearance	Growth (2nd month) upon early treatment discontinuation (1st month)
4	Overall negative increment. Positive increment detected after treatment discontinuation (1st month)	Flattening, decrease in erythema	Ulceration and growth (2nd month) upon early treatment discontinuation (1st month)
5	No temperature variation	Flattening, decrease in erythema	No
	Negative increment		
6	Negative increment	Flattening, total disappearance	No
7	Negative increment	Total disappearance	No
8	Negative increment	Decrease in erythema	No
9	Overall negative increment. Local increment detected upon transitory ulceration (5th month–7th month)	Flattening, decrease in erythema	Ulceration (5th month–7th month)
	Negative increment	Total disappearance	No

## Data Availability

The C++ source code developed to control the system and process image data is available in the following public Gihub repository: https://github.com/juanle82/Biomedical_IR_Project (accessed on 5 September 2022).

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
