# Peer review of "Thermography as a Method for Bedside Monitoring of Infantile Hemangiomas"

_cancers, 2022, doi:10.3390/cancers14215392_

Round 1
Reviewer 1 Report (Previous Reviewer 1)
good
Author Response
Dear Reviewer,
Find enclosed a Word document with the point by point responses to all your inquiries.

Reviewer 2 Report (Previous Reviewer 2)
1. In the Introduction: the phrase "Conventional diagnosis techniques to assess practitioners in determining the IH extension and localization" replace assess by assist.
2. Also "Among the vast spectrum of vascular anomalies, IHs are maybe the ones" replace are maybe by one of them.
3. The last 2 paragraphs of the introduction describe the study conclusion. The place of these 2 phrases is the conclusion or the discussion to summarize the results.
4. In methods, sections 2.2.2., and 2.2.3. till area estimation method, are preferably summarized especially section 2.2.3., which contains repeated information and phrases. The area estimation method, should be simplified to suit the medical and the electronic specialist readers. The conclusion, as well, can be summarized.
Author Response
Dear Reviewer,
Find enclosed a Word document with the point by point responses to all your inquiries.

Reviewer 3 Report (Previous Reviewer 3)
Authors have updated their manuscript to address issues and suggestions made, but there are still too many weaknesses to be convincing.
As I mentioned on my first report:
The linear link between blood flow and hemangioma evolution is not supported by enough scientific data and literature and should be studied and discussed with strong demonstration.
As requested, an example of a difficult to diagnose hemangioma has been added, but the technic is affected by too many factors such as hair and skin folds, as much as visual inspection. The conclusions about the technic are not supported by the data.
Author Response
Dear Reviewer,
Find enclosed a Word document with the point by point responses to all your inquiries.

Reviewer 4 Report (Previous Reviewer 4)
This paper on the interesting method of thermography for assessment of infantile hemangiomas (IH) using a customized, handheld device has clearly improved by the revision process. Still, several sections are very repetitive (e.g. reiteration of the study setup) and the manuscript is overly verbose, often making it hard for the reader to catch the authors’ message. This would very much benefit from professional English language editing to make it more precise and concise and improve the flow.
Title:
Suggest adapting the title to something more specific such as “Thermography as a method for assessing bedside monitoring of infantile hemangiomas” The authors could also consider integrating that the results are in line with that of dermatologist.
Simple summary:
- I doubt that the average layperson can understand the concept of “clonal expansion”.
Abstract
- Line 24: “For a correct evolution, […]” This is incorrect wording. What the authors probably mean is: “To predict the clinical course and counsel on treatment, it is crucial to 24 determine precisely its extension, volume, and location.”
- Lines 26-27: “Diagnosis is commonly based on area, texture, and color estimations in conjunction 27 with Doppler Ultrasonography or Magnetic Resonance.” I disagree. Diagnosis of infantile hemangiomas is made by clinical inspection and palpation. Only in some cases, mostly to measure depth, additional imaging is needed. Please rephrase.
Introduction
- Line 49: Ref [1] is incorrect here.
- Line 52-53: “Conventional diagnosis techniques to assess practitioners in determining the IH ex-52 tension and localization are Doppler Ultrasonography and Magnetic Resonance Imaging (MRI).“ Probably the authors mean “assist” instead of “assess”.
- Line 56: “Moreover, they need the use of gels and touching the patient, who is usually an infant, reluctant to them.“ This is an awkward sentence, please delete or rephrase.
- Lines 72-73: “[…] thermography as a tool to foresee the correct hemangioma evolution without treatment“ I think what the authors mean here is “to correctly foresee the evolution of a hemangioma”
Materials and Methods
- Thank you for adding more detail regarding the components of the device. These are, however, still missing specification and company names/locations. Is a patent for this in place?
- The section has benefited from the added information on programming. Will the authors share the code somewhere, e.g. at github?
- Lines 214-215: “Thermography proves to be a very competitive tool to monitor the IH extension before and during treatment.” And 217-219: “To assess the IH evolution during treatment, area estimations are a powerful tool. As 217 will be discussed in Section 3.4, summarizing the results of a pilot study, there is a corre-218 spondence between IH extension and the correct evolution.“ These statements cannot be made in the methods section, which is supposed to describe the method, not discuss the result.
Results
- Lines 262-264 “If the evolution of the IH after treatment is correct, the temperature difference between the histogram modes should be minimized.“ This is a hypothesis – as such, the place for this would be the introduction. The results section should either prove or debunk this.
- Figure 3 and 4: The age of the patients has been added to the text. It would be better to put it in the figure legends, so that the reader can understand and assess the figures without having to search through the text.
- Figure 5 has definitely improved by merging the two figures!
- Lines 308-309: “However, the thermal 308 image provides more accurate information about the malformation extension“ and figure legend in line 314: An IH is not really a malformation, consider changing the wording here.
- Pilot study lines 316-320: This is a repetition from the methods sections and has to be deleted here.
- Could the authors add a flowchart for the different study parts to illustrate this, and add patient numbers in there? This would be more useful than to reiterate the study setup again and again.
- Line 342: “a multifocal cutaneous hemangioma“ This may be nitty-gritty, but the correct term is “multifocal cutaneous hemangiomas”
-
Discussion
- The discussion section has improved. It still needs revision for conciseness and better flow of argument. There is still a lot of repetition to the previous parts, and although the authors provide more comparison and discussion with regard to other publications in this revised version, there is still room for improvement with regard to this.
- Line 399: “to assess the patient’s evolution” This is probably meant to be the hemangioma evolution, not the patient’s
- Line 415-416: “These ancillary techniques are not always straightforward during the clinical session because they require advanced equipment,gels, and trained operators. Children may be apprehensive about gels, general anesthesia, and the procedure itself.“ This is way too repetitive to the introduction!
Conclusions:
- Lines 450-457: This is another summary, not conclusion, and needs to be deleted from this section.
Data availability statement:
- Will the authors share the code somewhere, e.g. at github?
Author Response
Dear Reviewer,
Find enclosed a Word document with the point by point responses to all your inquiries.

Round 2
Reviewer 3 Report (Previous Reviewer 3)
Authors have updated their manuscript to address issues and suggestions made, but there are still too many weaknesses to be convincing.
As I mentioned on my first report:
The linear link between blood flow and hemangioma evolution is not supported by enough scientific data and literature and should be studied and discussed with strong demonstration.
The agreement with doppler blood flow measurement is often mentioned in the manuscript but never actually compared with presented data. So, the correlation between temperature variation and blow flow alteration cannot be evaluated. It is especially important as the correlation between blow flow and response to treatment is still in debate (Parapatt et al., 2021)
Parapatt, G. K., Oranges, T., Paolantonio, G., Ravà, L., Giancristoforo, S., Diociaiuti, A., Hachem, M. el, & Rollo, M. (2021). Color Doppler Evaluation of Arterial Resistive Index in Infantile Hemangioma: A Useful Parameter to Monitor the Response to Oral Propranolol? Frontiers in Pediatrics, 9, 1369. https://doi.org/10.3389/FPED.2021.718135/BIBTEX
Author Response
Dear Reviewer,
Find enclosed a Word document with the answers to your inquiries.
Best.

Reviewer 4 Report (Previous Reviewer 4)
The manuscript has further improved and all my comments and questions were adequately addressed.
One last thing: Thank you for adapting the title. I realized that in my suggestion to modify it, there was one word too much. It should better be “Thermography as a method for bedside monitoring of infantile hemangiomas”.
Author Response
Dear Reviewer,
Find enclosed a Word document with the answers to your inquiries.
Best.

Round 3
Reviewer 3 Report (Previous Reviewer 3)
Authors have updated their manuscript to address every issues and suggestions made.
This manuscript is a resubmission of an earlier submission. The following is a list of the peer review reports and author responses from that submission.
Round 1
Reviewer 1 Report
This is an exciting research paper.
However, a few suggestions are placed to further improve the manuscript.
Introduction:
Comment 1: It is better to be specific than generalised, ‘high incidence in children’.
Method:
Comment 2: Please ilaborate on, Inclusion/ Exclusion criteria.
Comment 3: Why were only Nine patients eligible for second phase?
Comment 3: Was any kind of medicine/ therapy received by the patients during two years? If yes, elaborate a bit.
Results:
Comment 4: ‘3.1.1. Image Segmentation’, firt half may be shifted to method section.
Comment 5: 3.1.2. Temperature Contour lines and Area Calculations, firt half may be shifted to method section.
Comment 6: ‘Several patients were recruited to participate in a two-year pilot study’, the context is not clesr! How much is several here?
Discussion:
Comment 7: The authors can elaborate a little more on their findings with respect to the previous research in discussion.
Author Response
Dear Sir/Madam,
Find enclosed a Word file with the answers to the questions you raise.
Best.

Reviewer 2 Report
The study is written well, with good presentation in most parts. Some aspects must be improved:
- The results indicate that 55 cases were studied. These 55 patients were only screened once in the first phase of the study. While the treated and followed up patients in the pilot study were only 9. This should be clearly described in patients and methods and in the abstract. The conclusion and the bulk of the results are based mainly on the pilot study findings
- The used term correlation between treatment effect and thermographic response appears as it carries a statistical meaning. Actually the findings are not statistically studied; that is why this should be described as a connection, association, or other related non-statistical term.
- The discussion is merely a repetition of what was mentioned in the Introduction and methods, no description or comparison with previous similar or related studies, no mention of the study shortcomings, and no discussion of the present study findings.
- The conclusion should not overestimate the results being a pilot study of 9 cases. Solid conclusions should not be drawn from a small preliminary study.
Author Response

(The authors gave the same response as above.)

Reviewer 3 Report
This study investigates the putative use of thermography to monitor infantile hemangioma diagnosis, growth phases and response to treatment such as betablockers.
This was done by measuring with a custom system based on IR the body surface temperature in order to translate in absolute temperature of the tumor.
The design and conclusions of this study are intriguing and potentially important for human health. Issues and suggestions for improvement are as follows:
Lines 14-15: sentence is not scientifically (and probably grammatically) correct: “The blood flow inside the tumor is higher due to the proliferation of endothelial cells in the blood vessels inside it.” Please revise according to literature. It is a general weakness for the study.
Lines 39-40: “Infantile hemangiomas (IH) are benign tumors (…) with a high incidence in children under the age of one year” this sentence is misleading, please correct: appear at birth or within the first few weeks after birth.
Lines 43, 139, 209: The technic is affected by hair as much as visual inspection, it should be mentioned clearly.
Lines 257-259: “hemangioma subtype and proliferation phase” the different hemangioma subtypes and the different phases of IH are not introduced which is unfortunate for non-specialized readers. Speaking of which including a different hemangioma subtype in the study as a control would have been interesting.
Figures 3 & 4: An example of a difficult to diagnose or a treatment response difficult to evaluate visually would have strengthen the study and convince of its clinical interest.
General remarks:
1) All IH have a fast growing phase in the first months of life followed by a slow involution. Those phases are easy for the dermatologist to identify, as it is subcutaneous, liver hemangiomas for instance would be some that need imaging follow-up for diagnosis and treatment evaluation.
2) Assessment of temperature in other subcutaneous hemangioma of infants such as non-involutive hemangiomas would have been informative.
3) This technic would be interesting if it could predicting relapses or ulcerations.
4) There is a lack of references about IH growing phases, biology and blood flow.
I would like to thank the authors and editors for the opportunity to review this interesting study
Author Response

(The authors gave the same response as above.)

Reviewer 4 Report
Leñero-Bardallo et al present an interesting study on the use of thermography for size measurement of infantile hemangioma, the most common benign skin tumor in infants. Their custom-made device is handheld and, according to the paper, provides direct (bedside) visualization of the extent of this vascular tumor. The thermographic images are easy to grasp and nicely presented with the corresponding clinical photographs. While this is a fascinating and potentially helpful method, the paper in its current form stays rather superficial and will not allow the reader to rebuild the experimental setup. The introduction stays vague, the methods lack precision, and the organization of the results section is at times confusing. The discussion is nearly non-existent (this section mostly copies the abstract) and does not discuss the author’s findings in context of any literature.
Simple summary and abstract:
- These are currently pretty similar. The abstract needs to be more specific and should be aimed at a scientific audience.
- One main problem with MRI in infants is the need for general anesthesia.
Introduction:
- This section could be improved by defining the problem (accurate size measurement of IH), the current approaches to it, and the method of thermography and its potential use for IH more clearly. It would be also helpful to mention the characteristic growth pattern of IH and different IH types here, and to say for which of these stages/types the author envision thermography to be helpful. In lines 65-76, the study objectives should be stated more precisely.
- Line 41: The majority of IHs do not require any treatment! Propranolol has only been introduced to IH therapy a decade ago (but has without question revolutionized its therapy) and requires references here.
- Line 48-51: MRI usually requires anesthesia. Adapt wording to not copy the abstract.
Materials and methods:
- Lines 78-83: This should go into the introduction.
- Lines 84-95: Naming of parts and manufacturers needed to allow readers to rebuild the experimental setup.
- Lines 96-97: Time period (month/year), inclusion/exclusion criteria (patient age? IH growth phase?) needed.
- Figure 2: How old is this child? He looks like a kindergardener, which would be rather old to receive IH therapy.
- Which program and methods were used for programming the device and analyses?
Results
- Throughout the paper, the word “therapy evolution” is used. This is not a common term and does not make sense to me. Please review this carefully and replace.
- Figures 3 and 4: How old are the children in the photos?
- Lines 208-210: Repetitive to previous sections.
- Figures 5 and 6: It looks like this is the same child shown in Figure 3. This should be pointed out. As the thermographic image in 3b, 5a, and 6 a are the same, it might be helpful to integrate these into one figure.
- Figure 6b and 6c: Why do the same lines have different colors in the overview and close up sections?
- Lines 244-247: Will the system convert the measurement into cm2?
- Line 249: “Several patients were recruited” is very imprecise. Patient characteristics should be provided earlier (age range, sex ratio, IH type and growth phase – this is currently only available for the 9 patients studied longitudinally, but not for the 55 patients. For the 9 patients, the information should be provided earlier).
- Line 259-260: The definition of the proliferation phase is not in line with definitions in literature.
- Line 271, “multiple cutaneous hemangiomatosis”: Revise wording – either cutaneous hemangiomatosis or multifocal cutaneous hemangioma.
- Table 3: Column “IH type”: These types should be explained to readers and reference given. Column “Localization and dimensions”: Is this determined by the thermography analysis or visually?
- Table 4: “Response parameters” should rather be “Clinical response”. For “Incidence and side effects”, cases 1 and 4 lack the timeframe for early treatment discontinuation and regrowth. Case 3: “Nervousness” seems and inappropriate word here. Case 9: At what timepoint did ulceration occur, when did it resolve, and what did thermography show?
- Lines 309-316: This is not results.
Discussion:
- Very repetitive to the abstracts, too superficial, needs actual discussion in light of the existing literature.
Data availability statement: I do not see the data as a supplementary material. Was this missed to upload?
References:
- References 12-14 appear incomplete, please double check.
Author Response

(The authors gave the same response as above.)
